# A Post-Operative Follow-Up of an Endangered Saltwater Fish Lensectomy for Cataract Management in a Public Aquarium: A Case Series

**DOI:** 10.3390/vetsci10100611

**Published:** 2023-10-08

**Authors:** Julie Pujol, Benjamin Lamglait, Maria Vanore, Catherine Rousseau, Claire Vergneau-Grosset

**Affiliations:** 1Département de Sciences Cliniques, Faculty of Veterinary Medicine, Université de Montréal, 3200 Rue Sicotte, Saint-Hyacinthe, QC J2S 2M2, Canada; julie.pujol@umontreal.ca (J.P.); maria.vanore@umontreal.ca (M.V.); 2Centre Québécois sur la Santé des Animaux Sauvages, 3200 Rue Sicotte, Saint-Hyacinthe, QC J2S 2M2, Canada; benjamin.lamglait@zoobeauval.com; 3Aquarium du Québec, 1675 Avenue des Hôtels, Québec, QC G1W 4S3, Canada; rousseau.catherine@sepaq.com

**Keywords:** lens extraction, outcome, piscine, striped bass, wolffish

## Abstract

**Simple Summary:**

Ocular pathologies, such as cataracts, may affect fish welfare. In severe cataract cases, complications such as anorexia, low body condition score, skin wounds, and scoliosis from chronic abnormal posture may develop in fish. Ultimately, cataracts may be associated with an inability to feed, cachexia, and death. The major surgical treatment of cataracts in fish is lens extraction. Although several cases have been published, little is known regarding the outcome and long-term prognosis of fish undergoing lens extraction. Eleven fish housed in a public aquarium had their lenses extracted after mature cataracts were diagnosed. As lens extraction was bilateral in four cases, follow-ups were available for 15 eyes. Following lens extraction, 73% of the fish resumed feeding, regained a normal body condition score, and appeared clinically normal. Surgical technique and post-operative complications are further detailed in this article. This study reviews the clinical outcome of lens extraction and could help clinicians improve the management of cataracts in fish.

**Abstract:**

Mature cataracts can be a life-threatening condition in fish as it may result in anorexia. Lens extraction has been previously described in fish, but the long-term outcome of this procedure has not been evaluated. Eleven captive-bred adult fish housed in a public aquarium presented with unilateral (*n* = 3/11) or bilateral (*n* = 8/11) mature cataracts. All cases belonged to three endangered fish populations: striped bass (*Morone saxatilis*) originating from the Saint Lawrence River and Atlantic and spotted wolffish (*Anarhichas lupus* and *Anarhichas minor*). Pre-operatively, fish presented with anorexia, dark discoloration, abnormal position in the water column, skin abrasions, and/or decreased body condition scores. A lensectomy was performed in eleven fish, including a bilateral procedure in four fish, corresponding to fifteen eyes. Follow-up examinations were performed one to two weeks and one year post surgery. The main complication was self-resolving: corneal edema was present in 67% of eyes, persistent after a week in 47% of eyes, and resolved thereafter. Post-operatively, 73% of fish resumed feeding and regained a good body condition score. Persistent visual impairment post-surgery associated with anorexia required euthanasia in three out of eleven cases. The median time of death was 336 days post surgery (range: 27–1439 days) and three cases were still alive 1334 to 1425 days after the lensectomy. This was considered a favorable outcome for these older individuals with concurrent diseases.

## 1. Introduction

A cataract is defined as an opacity of the lens and is a common condition in fish [1,2]. The incidence of cataracts has been reported to occur in fish secondary to osmotic and nutritional imbalances, suboptimal temperature, excessive ultraviolet radiation, intraocular parasites, toxins, trauma, and genetic predisposition [1,3]. In comparison with other species, cataracts secondary to age-related lens degeneration are not reported in fish. The overall prevalence of cataracts in fish is not known, and is highly dependent on fish population and etiology. It has been reported as high as 100% in wild and farm-raised lumpfish [4]. In wild fish, cataracts are often secondary to the parasite *Diplostomum*. However, this condition is less developed in closed environments, as the parasite cycle requires a bird as definitive host and a snail as intermediate host [1].

Cataracts can lead to blindness. In general, blind fish are able to avoid obstacles using their lateral line but they may experience trauma from interactions with conspecifics [3]. Moreover, the ability of feeding can decrease as many teleost fish need to be exposed to light to feed [5]. Thus, mature cataracts can directly impact fish welfare and can be life-threatening due to causing anorexia [1,6]. Cataract treatment is therefore indicated to restore food intake and avoid complications in untreated cataracts, such as uveitis [2]. For treatment, acute osmotic cataracts may be reversible with adequate salinity [1]. For other etiologies, cataract reversibility has not been reported with husbandry-related changes or medical treatment. Therefore, surgical lensectomy, as described elsewhere [7], is the preferred treatment for fish cataracts. Phacoemulsification is used on cataracts secondary to intraocular parasites that alter the lens nucleus’ structure [2]. To the authors’ knowledge, the long-term post-operative outcome of fish lensectomy has not been reported.

The three species included in this case series are from endangered fish populations. The Saint Lawrence River striped bass (*Morone saxatilis*) used to be extinct in the wild and was reintroduced through a recovery program [8]. The spotted wolffish (*Anarhichas minor*) is a threatened species [9], while the Atlantic wolffish (*Anarhichas lupus*) is a species of special concern [10]. Due to the population status of these species, a high level of veterinary care is expected to increase their life expectancy under managed care. The first aim of this study was to provide care for these endangered species and to treat cataracts in order to improve welfare and cure this life-threatening condition. The second objective was to report the outcome of lensectomy procedures and the survival time after this procedure in these aquarium-housed fishes. We hypothesized that lensectomy in fish would result in minimal long-term complication and that can improve welfare and food intake in this fish population.

## 2. Materials and Methods

### 2.1. Fish Population

Eleven fish from the Aquarium du Québec were evaluated on-site by the Zoological Medicine service of the Université de Montréal between 2015 and 2021. All individuals were adult striped bass or Atlantic or spotted wolffish presenting with mature cataracts and were over 10 years old based on their size and date of purchase from local farms [11,12].

Nineteen striped bass were initially kept in three indoor multispecies exhibits: a 44,600-L saltwater exhibit kept at 7 °C, a 350,000-L saltwater exhibit kept at 12 °C, and a 25,000-L freshwater exhibit kept between 17 and 19 °C. In 2016, all striped bass were transferred to the 350,000-L saltwater exhibit at 13 °C. Before 2016, lights above the habitats included fluorescent tubes and metal halide lights (Ultra-arc 8000 lumens, United States Radium Corporation, Orange, NJ, USA). After 2016, lights were replaced with Light-Emitting Diodes (Cannon Pro LED 120 W, Ecoxotic, Vista, CA, USA). New cases continued to be reported after the change in lights.

A total of 10 Atlantic and 12 spotted wolffish were kept in two indoor saltwater multispecies exhibits of 106,000 L and 44,600 L at 7 °C. These habitats were equipped with fluorescent lights of low intensity to mimic a deep-water environment.

The life support system of each of these exhibits included a biofilter and/or sand filter, an ultraviolet sterilizer (1-L-G or 2-L-G model) or an ozonation system, and a protein skimmer. Water parameters (pH, salinity, temperature, ammonia, nitrites) were checked once every two weeks. The salt used was Instant Ocean Sea salt (Reef crystals and salts, Instant Ocean, Blacksburg, VA, USA) or Crystal Sea Bioassay Marinemix salt (Marine Enterprises, Baltimore, MD, USA).

Fishes were fed twice weekly with fish and crustacean prey items supplemented with a commercial food (Mazuri Omnivore Aquatic Gel Diet, Land O’Lakes Inc., Arden Hills, MN, USA) and a mineral supplement (Mazuri Vita-Zu Bird Tablet, without Vit A, Land O’Lakes Inc., Arden Hills, MN, USA). After 2018, a pelleted diet (Vitalis Marine pellets, Vitalis Aquatic Nutrition, World Feeds Ltd., Thorne, UK) was also offered twice weekly.

### 2.2. General and Ophthalmic Examinations

Vision was evaluated based on appetite, body condition score, skin color, repeated traumas or secondary wounds, and the absence of avoidance behavior when a net gently approached one side of the fish. Whenever a fish displayed clinical signs suggestive of impaired vision, it was caught and a complete physical examination including an ophthalmic examination was performed under immersion in tricaine methanesulfonate (MS-222, Aqualife, Syndel, Nanaimo, BC, Canada) at 90–100 mg/L. The eyes of these species were around 20 mm in diameter. An ophthalmologic examination was carried out using a direct ophthalmoscope (3.5 V AutoStep Coaxial Ophthalmoscope, Welch Allyn Canada Ltd., Mississauga, ON, Canada). The aqueous flare could not be clearly evaluated using a direct ophthalmoscope. If needed, an ocular ultrasound was performed using an ophthalmic probe (13–6 MHz) (Sonosite Edge II Ultrasound, Fujifilm HealthCare, Lexington, MA, USA). Intra-ocular pressure was measured using rebound tonometry (Tonovet, Icare, Vantaa, Finland). Electroretinography was not available on-site and was therefore not performed. Fish were individually marked during their initial examination using a subcutaneous visible implant elastomer tag (Northwest Marine Technology, Anacortes, WA, USA) to enable subsequent follow-ups.

### 2.3. Surgical Procedure

When vision was altered, with an impact on general condition, a lensectomy was elected and performed by diplomates of the American College of Zoological Medicine. Anesthetic induction was performed via immersion in MS-222 at 90–100 mg/L. Florfenicol (Nuflor, Merk Animal Health, Kirkland, QC, Canada) 30 mg/kg intramuscularly and butorphanol (Torbugesic, Zoetis Canada Inc., Kirkland, QC, Canada) 0.4 mg/kg intramuscularly [13] were administrated pre-operatively. The fish was placed in lateral recumbency on a surgical table designed for the continuous irrigation of the gills [14]. A maintenance concentration of MS-222 was adapted depending on anesthetic depth. Water temperature was monitored and maintained at 13 °C with ice packs. All procedures were performed by specialists in Zoological Medicine wearing surgical loupes (Designs of Vision Inc., Bohemia, NY, USA) with a magnification of ×2.5.

The eye was gently flushed with 60 mL of saline solution. Two drops of proparacaine (Alcaine 0.5%, Sandoz Pharmaceuticals, Princeton, NJ, USA) were instilled on the cornea [2]. Five minutes later, two drops of either tropicamide (Mydriacyl, Alcon Laboratories Inc., Mississauga, ON, Canada), ophthalmic atropine (Isopto atropine, Alcon Laboratories Inc., Mississauga, ON, Canada), or rocuronium bromide (Pfizer Canada ULC, Kirkland, QC, Canada) were instilled. A clear surgical drape (VSP transparent adhesive surgical drape, Millennium Surgical Corp, Bala Cynwyd, PA, USA) was placed on the fish. The bulbar conjunctiva, 2 mm from the dorsal limbus, was immobilized using Bishop-Hartmann forceps and then the perilimbal cornea was incised with a keratome or a number 12 surgical blade. Viscoelastic (Biovisc 1.2% Hyaluronic acid, Acrivet Inc., Salt Lake City, UT, USA) was injected into the anterior chamber, or between the cornea and the lens, for cases with anterior lens luxation, in order to protect the endothelium. Incision was prolonged over 120° using curved sharp iris scissors (Figure 1). The sharp iris scissors were then inserted between the iris and the lens to sever the ventral *retractor lentis* muscle from the lens and dissect the zonular fibers over 360°. The lens was gently caught with Adson tissue pliers and was extracted through the corneal incision. The corneal incision was closed using a resorbable 5–0 or 6–0 polydioxanone monofilament (PDS, Ethicon, Somerville, NJ, USA) using a simple interrupted suture pattern (Figure 1). Remaining gas bubbles in the anterior chamber, if present, were aspirated before suturing. The Seidel test was performed to evaluate the aqueous humor leakage from the anterior chamber, assessing the suture tightening.

Post-operatively, meloxicam (Metacam, Boerhinger Ingelheim Vetmedica, Duluth, GA, USA) at 0.5 mg/kg or robenacoxib (Onsior, Elanco US Inc., Greenfiled, IN, USA) at 2 mg/kg were injected in the epaxial muscles [15].

### 2.4. Post-Operative Follow-Up

The administration of anti-inflammatories was repeated 3 to 6 days after surgery during the first re-evaluation. Then, fishes were examined 7–14 days after surgery. Ophthalmic reassessment included intra-ocular pressure measurement if uveitis or glaucoma were suspected based on the presence of an aqueous flare or suspected buphthalmos. All absorbable sutures were resorbed spontaneously without complications. Re-evaluations were grouped into two categories: one made between day 15 and day 365, and one after day 365. Each included a general and ophthalmic examination under general anesthesia.

In case of death, fish were necropsied, and the eyes and other organs were submitted to the Centre Québécois sur la Santé des Animaux Sauvages (CQSAS) for histopathological analysis. Ocular histopathology was only submitted if abnormal vision was suspected by the attending veterinarian. Tissues were fixed in 10% neutral buffered formalin. After fixation, samples were routinely processed, embedded in paraffin, trimmed, and stained with hematoxylin and eosin.

### 2.5. Statistics

A Kaplan–Meier curve was produced for this group of fish in which lensectomy was performed (R version 4.0.2, R Core Team, Vienna, VA, Austria).

## 3. Results

### 3.1. Study Population

The study population is presented in Table 1. Eleven fish, including *M. saxatilis* (*n* = 7), *A. lupus* (*n* = 2), and *A. minor* (*n* = 2) with a median body weight of 1.6 kg (range: 1.3–6.5 kg), were included. The incidence of fish exhibiting signs of impaired vision attributable to cataracts was as follows: 37% (7/19) for striped bass, 20% (2/10) for Atlantic wolffish, and 17% (2/12) for spotted wolffish.

### 3.2. Pre-Surgical Clinical Signs

Prior to surgery, the most common manifestations of visual impairment included: anorexia or dysorexia (11/11, 100% fish), dark body color (4/11, 36% fish), skin abrasions (4/11, 36% fish), a low body condition score (2/11, 18% fish), and abnormal position in the water column (1/11, 9% fish).

Two fish had anteriorly luxated lenses associated with mature cataracts on presentation. In all other cases, the exclusive ophthalmologic finding was a unilateral (*n* = 3/11) or bilateral (*n* = 8/11) mature cataract. Pre-operative intraocular pressure and other ocular ultrasound findings were within normal limits (median intraocular pressure = 8.5 mmHg) [16].

### 3.3. Lensectomy

Animals underwent a unilateral (*n* = 7/11) or bilateral (*n* = 4/11) lensectomy. In one case, a bilateral lensectomy was performed on the same day. In the other three cases, the lensectomy of the second eye was performed secondarily to allow for the healing of the first eye before surgery on the contralateral eye.

All 11 fish recovered well from surgery. Surgery lasted 30 to 60 min per eye. The only anesthetic complication was the occasional mild movement of the fish during corneal incision that resulted in mild hyphema in five fish. A peri-operative complication observed occasionally was posterior lens luxation if the *retractor lentis* muscle was incompletely transected. Otherwise, the lens was retrieved without any peri-operative complication.

### 3.4. Outcome

The case progression is presented in Appendix A. The final outcome was considered favorable in 73% (8/11) of fish, which resumed feeding and could be returned to their exhibit. Persistent dark color, anorexia, and decreased body score were reported in three fish despite the lensectomy (3/11, 27%). Transient corneal edema was noted 3 to 15 days after surgery in 10 eyes (10/15, 67%). Other transient complications occurring mainly in the first 15 days post surgery included hyphema (5/15, 33%), gas bubbles (4/15, 27%) in the anterior chamber, uveitis (7/15, 47%), and iris oval distortion (1/15, 7%). When gas bubbles were large, fine-needle aspiration was performed. This procedure of transcorneal gas aspiration was performed in one fish without adverse effects.

Corneal opacity or corneal fibrosis was reported if the suture line was within 2 mm of the limbus and occurred in 47% (7/15) of eyes after day 15. Posterior and anterior synechiae developed in 47% (7/15) cases (Figure 2). Retinal detachment (4/15, 27%) (Figure 2) was diagnosed using ultrasound in three fish during follow-up examinations at 66, 98, and 312 days post surgery. The last retinal detachment was identified more than one year after lensectomy in post mortem analysis. Other late post-operative complications included enophthalmos (2/15, 13%) and mild buphthalmos (2/15, 13%) without increased intraocular pressure. A retro-orbital lipoma developed in one fish three years after lensectomy and was considered unrelated to the procedure. Scoliosis developed in two fish (2/11, 18%) in which the contralateral eye was affected.

Post mortem examinations are available in eight out of eleven fish to date. The three remaining cases are still alive. The cause of death was determined to be unrelated to the lensectomy in five out of eight cases (Appendix A). The histopathology of the eye and other organs are available in three out of eight cases (Appendix A), revealing a retinal detachment in all three cases.

The median time of death was 336 days post operation (range: 27–1439 days). The Kaplan–Meier curve is reported in Figure 3.

## 4. Discussion

This study described the surgical results of lensectomies on fish species that have not been reported previously. Overall, 73% (8/11 cases) of fish having undergone a lensectomy resumed feeding, maintained a good body condition, and had a sufficiently esthetic appearance to return to their exhibit. The median time of survival was 336 days, but certain cases of this case series have lived more than 3 years post lensectomy. The exact age of each fish was not known, but all were over 10 years old and considered to be of middle to advanced age, knowing that the maximum age reported was 30 years old [17]. Some had other comorbidities, including germ cell neoplasia, xanthomatosis, and nephrocalcinosis [10,18]. Despite complications resulting in unfavorable outcomes in 3 of 11 fish, this surgery could be recommended as a treatment for cataracts in fish. These three fishes remained blind and anorexic post operation. It is unknown whether the loss of vision was due to the procedure itself or to a retinal lesion present prior to the lensectomy. Ideally, a pre-operative ocular ultrasound and electroretinogram should have been performed in all cases. While the use of electroretinograms has been described in fish [19], this equipment was not available on-site. Due to the stress and significant risk of transport-related trauma in large fish, it was decided not to perform electroretinograms in these cases and instead to follow their post-operative evolution. In addition, cataract removal would have reduced the risk of secondary uveitis. If future cases were to be performed, an electroretinogram would be recommended prior to surgery to better select surgical candidates.

Assessing the efficacy of a procedure in aquatic medicine is often complex, and the principles applied in human or companion animal medicine do not always translate directly, primarily due to the distinct environmental conditions in aquatic life. In the present study, the fish were subject to veterinary evaluation due to their anorexic behavior and manifestations of blindness. A comprehensive examination led to the diagnosis of cataracts. The primary objective of the surgical interventions was to mitigate the clinical signs, specifically anorexia and blindness. It is worth noting that feeding represents one of the five fundamental freedoms of animal welfare. In this context, the recommendation for cataract surgery remained contingent on its success in restoring the fish’s ability to feed and improving their overall well-being. For future research endeavors pertaining to fish cataracts, the inclusion of a non-treatment control group, contrasted with a treatment group undergoing lensectomies, could offer a comprehensive assessment of surgical impact. However, it is crucial to recognize that such an approach fell beyond the purview of this study, where the primary objective was to address the welfare of endangered fish afflicted by mature cataracts which were significantly affecting their quality of life.

The resurgence of appetite and the amelioration in overall condition have been tentatively ascribed to an augmentation in visual function subsequent to the surgical intervention. Given that a fish’s feeding capacity wanes due to light deprivation, it is plausible that surgery contributed to the restoration of their appetite [5]. Nonetheless, this correlation remains speculative. Additional factors may also exert influence, including the potential alleviation of discomfort following cataract treatment.

Accommodation refers to the eye’s ability to focus on objects at different distances. The accommodation of a fish’s eye is different from that of mammals due to the anatomy of its lens [20]. The accommodation of teleost is accomplished through the movement of the lens by contracting the *retractor lentis* muscle [21]. Despite having aphakic eyes post operation, fish were observed feeding and gained weight [5,22]. Overall, the clinical outcome observed in fish after a bilateral lensectomy suggests that fish can adapt with aphakic eyes. The insertion of an intraocular lens is well described in dogs [23], but has not been reported in fish to date. So, currently, there is no option to replace the fish’s crystalline lens after a lensectomy. A previous study has also shown that fish can adapt to unilateral vision after placement with a prosthetic eye after unilateral enucleation [24]. That previous study is encouraging in the field of research aimed at developing prosthetic materials in fish, including lens prosthesis.

Concerning post-operative complications, in this study, corneal edema was the most frequent complication observed in 67% (10/15) of eyes in the early post-operative period. This incidence rate is high, but similar results were reported in other species as a very common post-operative finding, with 100% of horses reported to develop edema after phacoemulsification [25]. Corneal edema could be attributed to an inflammatory reaction secondary to surgery or increased intraocular pressure [26]. Indeed, temporary post-operative ocular hypertension is commonly reported within 8 h in 16% of dogs after cataract surgery [27]. Intraocular hypertension was probably underestimated in the present study, as the first re-evaluation was performed after three days. The choice of suture material can also have an impact on post-operative inflammation. Synthetic monofilaments were used to decrease inflammation and bacterial contamination in this case series, as previously recommended for fish [28].

Retinal detachment was seen in four eyes (4/15, 27%). This complication rate is high, with an incidence rate of 1.8% described in horses after phacoemulsification [25] and 1–8.4% in dogs [29,30]. In humans, lensectomy increases the risk of retinal detachment compared to phacoemulsification [31]. Histologically, the pathologic retinal detachments described in fish have the same characteristics as those found in mammals with the significant multifocal vacuolation of the retinal tissue [32]; retinal folds, cell clumping, and metaplasia with pigment accumulation were also visible within the detached retinas. In addition, surgeries on mature cataracts could have increased the risk of retinal detachment [26]. Indeed, in dogs, advancing grade in vitreous degeneration is associated with the progression of cataracts, and vitreous degeneration is a complicating factor in retinal detachment [33]. Thus, in domestic animals, surgery is recommended in the early stages of cataracts to reduce complications linked with the development of retinal detachment and uveitis [30]. This recommendation was difficult to follow because fish were caught only when aquarium divers detected ocular abnormalities. However, preventive retinopexy or a different surgical technique to avoid the accidental posterior luxation of the lens might be considered in future cases. In particular, holding the lens via the insertion of a needle might be a future avenue to consider in fish.

In the present study, the incidence of hyphema was higher in fish (33%, 5/15) than reported in dogs (12.3%) [29]. It is likely that intraocular hemorrhage was of traumatic origin during surgery. Non-depolarizing neuromuscular blocking agents should be used in fish to induce a flaccid paralysis [34]. One case of endophthalmitis was reported in this study, while an incidence rate of 1.4% to 27% has been reported in dogs after cataract surgery [30,35]. Uveitis was reported in 47% of fish (7/15) during the first week post surgery. In comparison, this complication occurs in 16.2% of dogs at a median time of 85 days post-phacoemulsification [29]. In this study, the prevalence of uveitis may be underdiagnosed because all the fish were treated with systemic non-steroidal anti-inflammatory drugs.

The occurrence of scoliosis In two cases was suspected to be related to impaired unilateral vision following surgery. This skeletal lesion has been reported to develop in fish due to a postural adaptation to optimize the visual function in the case of visual deficiencies [22,36]. This suggests that a bilateral lensectomy should ideally be performed in cases with bilateral cataracts.

A definitive etiology of cataracts was not determined in this fish population. Some etiologies were ruled out, such as osmotic imbalance, suboptimal temperature, parasitic infection, and toxins. Exposure to ultraviolet radiation was considered as a factor. This was ruled out through the measurement of on-site ultraviolet emission with a spectrophotometer. An internal nutritional analysis study was conducted to determine whether the diets were adequate for striped bass and wolfish [37]. On average, choline intake was 1642% (1478–1793%) higher than the nutritional requirements. Choline excess could induce cataracts via an increased permeability of the lens [38]; thus, a nutritional etiology was suspected. The transition to a new diet based on balanced commercial pellets is ongoing in this aquarium. Finding the disease’s origin is crucial to preventing cataracts. Managing fish populations with cataracts should start with identifying the risk factors.

The limitations of this study included several factors such as the relatively small sample size of individuals and species under consideration. Future investigations should aim to encompass a more extensive range of fish species. The inclusion of three species may potentially obscure interspecies distinctions and introduce biases into the descriptions. As this study was retrospective in nature, certain medical records were found to have missing data. Furthermore, due to the involvement of different veterinarians in the assessment of the fish across the study, variations in case descriptions might have arisen, albeit all relying on terminology from conventional ophthalmological semiology rooted in comparative medicine.

## 5. Conclusions

Lensectomies improved the quality of life in the majority of fish cases reported in the present study. Following the retrospective analysis of these clinical cases, recommendations can be formulated. Performing a lensectomy at an earlier stage of cataract development may improve post-operative outcomes. The use of neuromuscular blocking agents should be evaluated in fish. Bilateral procedures should be attempted in the case of bilateral vision impairment to prevent scoliosis. Finally, the use of ocular ultrasound and electroretinogram assessment to detect pre-operative retinal detachment and functioning may help in selecting candidates and establishing the prognosis for vision recovery. Further studies on fish ophthalmology are needed to improve the description of their specific anatomy and physiology and the treatment of ophthalmologic problems.

## Figures and Tables

**Figure 1 vetsci-10-00611-f001:**
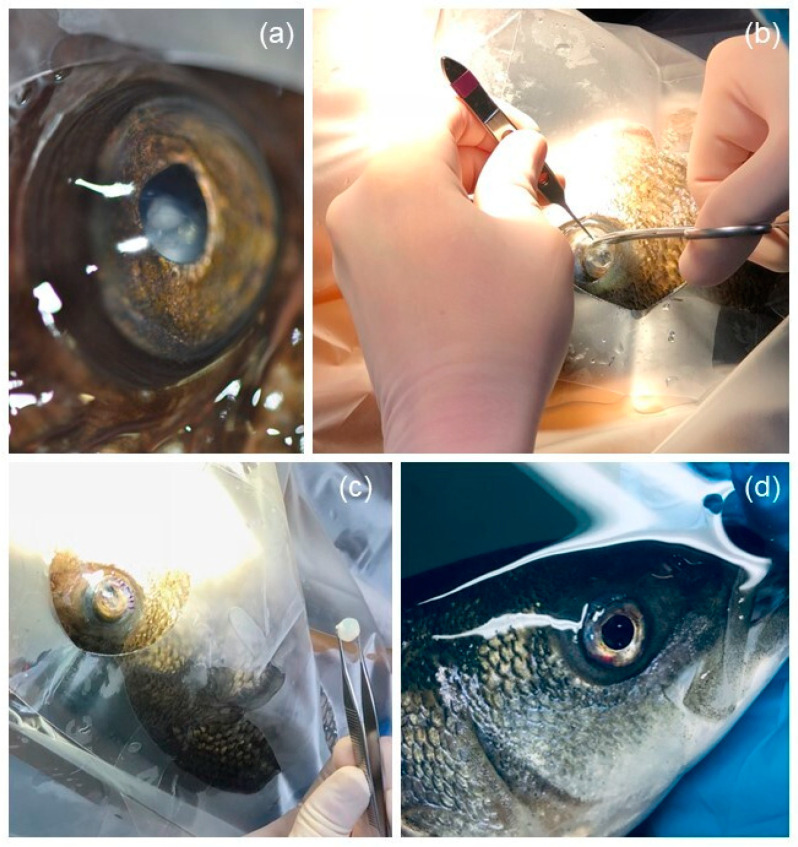
Steps for lensectomy performed on striped bass (*Morone saxatilis*), Atlantic wolffish (*Anarhichas lupus)*, and spotted wolffish (*Anarhichas minor)*. The eye diameter is around 20 mm. (**a**) Eye of a spotted wolffish before surgery presenting with cataract and ventral lens luxation. Note the characteristic cloudy aspect of the lens. (**b**) Incision of the ventrocaudal limbus using Bishop-Hartmann pliers and curved iris scissors. (**c**) Extracted lens. Note its large and spherical shape. (**d**) Closed limbus with simple interrupted sutures. Note the ventrocaudal hyphema.

**Figure 2 vetsci-10-00611-f002:**
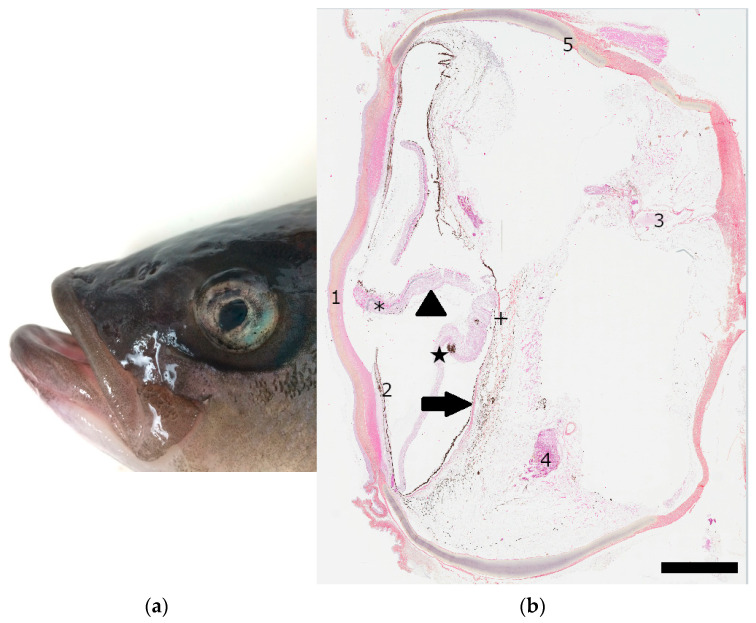
Examples of complications following lensectomy in striped bass (*Morone saxatilis*). (**a**) Photomicrograph of the eye of a striped bass during re-evaluation more than a year post-lensectomy: synechiae are noted. (**b**) Histologic image of the eye of a striped bass 315 days after lensectomy (hematoxylin and eosin staining, magnification ×20, scale bar = 2 mm), anterior synechiae and retinal detachment were present on antemortem ocular ultrasound. Retinal detachment is visible via the separation of the neurosensory retina (arrowhead) from the underlying retinal pigment epithelium (arrow) and the presence of localized hemorrhages with light lymphoplasmacytic infiltration (asterisk). Areas of retinal detachment were also identified histologically with the presence of retinal folds (cross). Clumping and metaplasia with pigment accumulation were visible within the detached retina (star). 1. Cornea. 2. Iris. 3. Optic nerve. 4. Choroid body (*rete mirabile).* 5. Sclera (with cartilage).

**Figure 3 vetsci-10-00611-f003:**
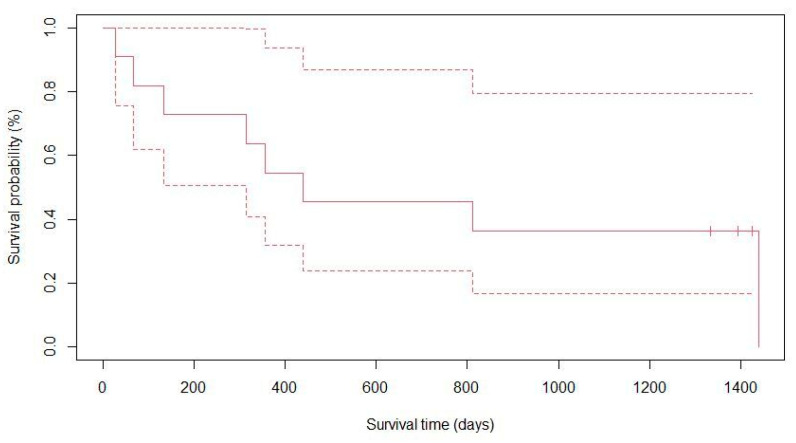
Kaplan–Meier solid curve including seven striped bass (*Morone saxatilis*), two Atlantic wolffish (*Anarhichas lupus*), and two spotted wolffish (*Anarhichas minor*) having undergone a lensectomy in a public aquarium. The dashed lines represent a 95% confidence interval.

**Table 1 vetsci-10-00611-t001:** Study population, sex distribution, body weight, and procedure performed in each individual.

Fish Number	Specie	Sex (Method of Determination)	Body Weight (kg)	Cataract	Bilateral Lensectomy
1	*Morone saxatilis*	Male (ultrasound)	6.5	Bilateral	No
2	*Morone saxatilis*	Male (gonadectomy)	1.4	Bilateral	Yes
3	*Morone saxatilis*	Male (necropsy)	1.7	Bilateral	No
4	*Morone saxatilis*	Unknown	1.7	Bilateral	Yes
5	*Morone saxatilis*	Male (gonadectomy)	1.6	Bilateral	No
6	*Morone saxatilis*	Male (necropsy)	1.5	Bilateral (one luxated lens)	Yes
7	*Morone saxatilis*	Unknown	1.3	Unilateral	No
8	*Anarhichas lupus*	Male (necropsy)	4.5	Bilateral	Yes
9	*Anarhichas lupus*	Unknown	1.5	Unilateral	No
10	*Anarhichas minor*	Male (necropsy)	2.5	Bilateral (one luxated lens)	No
11	*Anarhichas minor*	Unknown	1.4	Unilateral	No

## Data Availability

The data presented in this study are available in this manuscript.

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
