# Peer review of "A Post-Operative Follow-Up of an Endangered Saltwater Fish Lensectomy for Cataract Management in a Public Aquarium: A Case Series"

_vetsci, 2023, doi:10.3390/vetsci10100611_

Round 1

Reviewer 1 Report

See file

See file

Reviewer 2 Report

This manuscript is not well written, the ophthalmic procedures are not well described , some of them are missing . The bibliography is not updated . Limitations are not considered at al. The quality of pictures is very low.  Is important to indicate who performed the ophthalmic examination and  surgery .

some mistakes 

Reviewer 3 Report

The authors describe the surgical method of lensectomy in 3 different fish species in order to solve their problem of cataract. This solution seems very interesting because appears very efficient. The cataract in fishes is a serous pathology that can lead to death from anorexia. Postoperative complications are rare and this justifies this surgical procedure. It can be performed on highly valuable fishes present in aquariums. 

The manuscript is well written, conceived and documented. Introduction and material & methods are precise and informative. The results are convincing and supported by long term evaluation after the surgical procedure. 

I can't find any errors or details to correct. The unique think that the authors could add in the discussion is that the scientific team of the aquariums can adopt strategies to prevent the cataract in fishes (change in feeding, temperature control, water biophysical parameters ecc...). The surgical procedure of lensectomy is a good solution but it should be better prevent the cataract in fishes.     

Round 2

Reviewer 1 Report

Thank you for the replies. 

I agree with publication in the current form.

Author Response

The authors express their gratitude for the meticulous and constructive review of this manuscript.

Sincerely. 

Reviewer 3 Report

The manuscript has been now improved and I think that it deserves to be publisched in Veterinary Sciences. Best wishes 

Author Response

(The authors gave the same response as above.)
